# Ligand-modulated nickel-catalyzed regioselective silylalkylation of alkenes

Chao Ding[1,2], Yaoyu Ren[1,2], Yue Yu[1,2] & Guoyin Yin [1]✉

Organosilicon compounds have shown tremendous potential in drug discovery and their synthesis stimulates wide interest. Multicomponent cross-coupling of alkenes with silicon reagents is used to yield complex silicon-containing compounds from readily accessible feedstock chemicals but the reaction with simple alkenes remains challenging. Here, we report a regioselective silylalkylation of simple alkenes, which is enabled by using a stable Ni(II) salt and an inexpensive *trans*−1,2-diaminocyclohexane ligand as a catalyst. Remarkably, this reaction can tolerate a broad range of olefins bearing various functional groups, including alcohol, ester, amides and ethers, thus it allows for the efficient and selective assembly of a diverse range of bifunctional organosilicon building blocks from terminal alkenes, alkyl halides and the Suginome reagent. Moreover, an expedient synthetic route toward alpha-Lipoic acid has been developed by this methodology.

In drug discovery, the incorporation of a main group element functionality, such as silicon, into drug candidates is a promising chemical strategy that can improve their biological activity and reduce toxicity[1]. Silicon, which belongs to group IV in the periodic table, shares a comparable bonding structure with carbon. However, silicon has a larger covalent radius compared to carbon, leading to enhanced lipophilicity and distinct bonding preferences[2]. The C/Si replacement is a widely used structural modification strategy that enables the incorporation of silicon-containing functionalities into organic compounds, thus branching into new chemical space[3,4]. This approach significantly alters the molecular physicochemical properties and in vivo activity of drug candidates. For example, C/Si exchange in the molecule of 4-fluorophenylpropylpiperidine has shown different types of bioactivity[5] (Fig. 1a). The incorporation of sila-proline in a Pro-rich peptide (PPII) has been shown to increase cellular uptake by 20-fold and resistance to proteolytic degradation[6] (Fig. 1a). This approach has led to the development of a series of unnatural sila-amino acids that have been applied in the synthesis of therapeutically relevant compounds[7]. Moreover, the incorporation of the silyl amine group into indomethacin has significantly improved its bioactivity[8,9].

Along with the shift from flat targets to three-dimensional structures in medicinal discovery[10], the development of modular methods for the synthesis of sp³-rich organosilicon compounds has stimulated growing interest, leading to significant advances in the hydrosilylation of alkenes[11–14] and cross-coupling of alkyl coupling partners with silicon reagents[15–24]. Multicomponent reactions have shown advantages in terms of the abundance of starting materials and efficiency towards complex targets[25–27]. One such approach involves trapping the intermediates of silyl-metal[28,29] or silyl radical species[30–32] with alkenes to introduce another group (Fig. 1c). This approach requires the use of alkenes with adjacent π-bonds, benefiting the formation of π-benzyl or π-allyl stabilized metal intermediates to ensure reactivity. Recently non-activated olefins have also been explored, and the organoboron-catalyzed carbosilylation reactions involving silylium ions are particularly noteworthy[33,34]. Transition metal-catalyzed reactions were also developed, but the installation of a strong coordinating auxiliary group into alkenes plays a crucial role in the success of these protocols[35,36]. However, certain functional groups and certain carbon lengths are required, leading to limitations in the alkene scope[37,38]. In addition, a regioselective arylsilylation of terminal alkenes was achieved by nickel chain-walking catalysis, but this reaction scope is limited to alkenyl arenes[39]. This reflects the ongoing challenge of anchoring a metal catalyst at the carbon chain without the assistance of strong directing groups, as the carbosilylation of simple non-activated alkenes is rare[40].

[1]The Institute for Advanced Studies, Wuhan University, 430072 Wuhan, Hubei, People's Republic of China. [2]These authors contributed equally: Chao Ding, Yaoyu Ren, Yue Yu. ✉e-mail: yinguoyin@whu.edu.cn

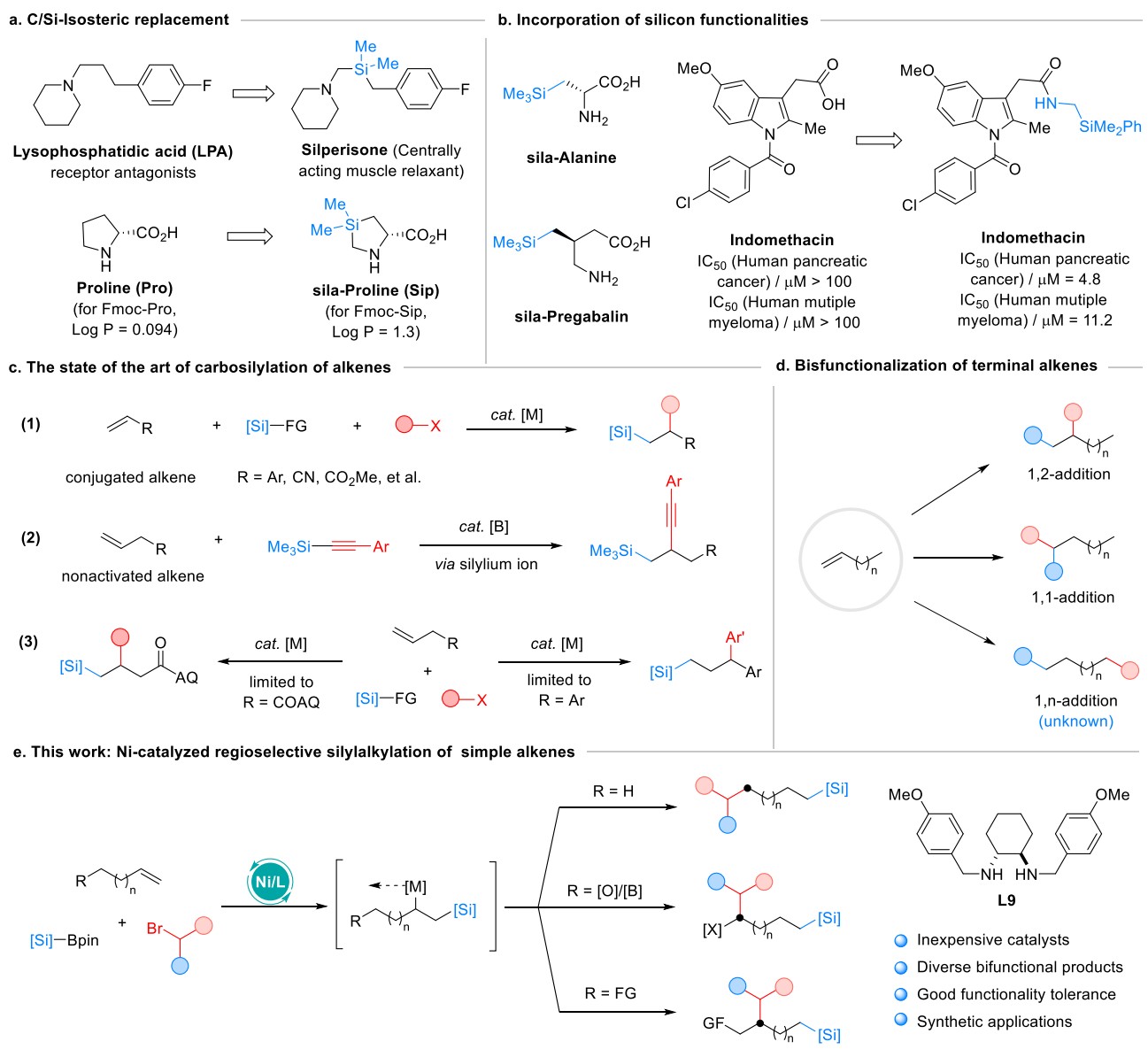

**Fig. 1 | Catalytic regioselective silylalkylation of alkenes. a** C/Si-Isosteric replacement; **b** Incorporation of silicon functionalities; **c** The state of the art of carbosilylation of alkenes; **d** Bisfunctionalization of terminal alkenes; **e** This work: Ni-catalyzed regioselective silylalkylation of simple alkenes.

On the other hand, alkene addition reactions through both 1,2-addition and 1,1-addition fashions have been widely recognized[41,42], while the addition reaction of simple aliphatic olefins in a highly selective head-tail addition is still underdeveloped (Fig. 1d). Motivated by our success in the borylative functionalization of alkenes by nickel chain-walking catalysis[43–45], we moved our interest to the silylative functionalization of alkenes. We hypothesized that if a highly regioselective silylative functionalization of simple unactivated alkenes was achieved, it would provide an efficient avenue for assembling organosilicon compounds from readily accessible materials. Herein, we present our successful development of a nickel-catalyzed regioselective silylcarbofunctionalization of simple alkenes, wherein a silyl group is introduced into terminal olefins concurrent with the construction of a C(sp³)-C(sp³) bond (Fig. 1e). The combination of a stable Ni(II) salt and the inexpensive *trans*−1,2-diaminocyclohexane enables different regioselectivities from different types of alkenes. This work not only provides an approach to enable access to diverse bifunctional building blocks from abundant materials but also demonstrates a

significant advancement in the chain-walking transformation of simple olefins.

## Results

### Reaction development

In the study, the Ni-catalyzed silylalkylation of an alkene was chosen as a model to investigate the feasibility of the speculation mentioned above. Commercially available 1-octene (**1a**), the Suginome reagent (**2a**)[46], and ethyl 4-bromobutyrate (**3a**) were used as model substrates. Ligand evaluation was conducted with NiBr₂·DME as catalyst, CuI as cocatalyst, and LiOMe as an activator for the Si-B reagent[47], in *N*-methylpyrrolidone (NMP) at 30 °C. It was found that the reaction chemoselectivity was greatly influenced by the nature of the used ligand. As shown in Fig. 2a, the ligands based on either 2,2'-bipyridine (**L1**), 1,10-phenanthroline (**L2**), bisoxazoline (**L3**), or 1,2-diphenylethylene-diamine (**L4**) backbone failed to produce any three-component coupling products. In contrast, the ligands based on the

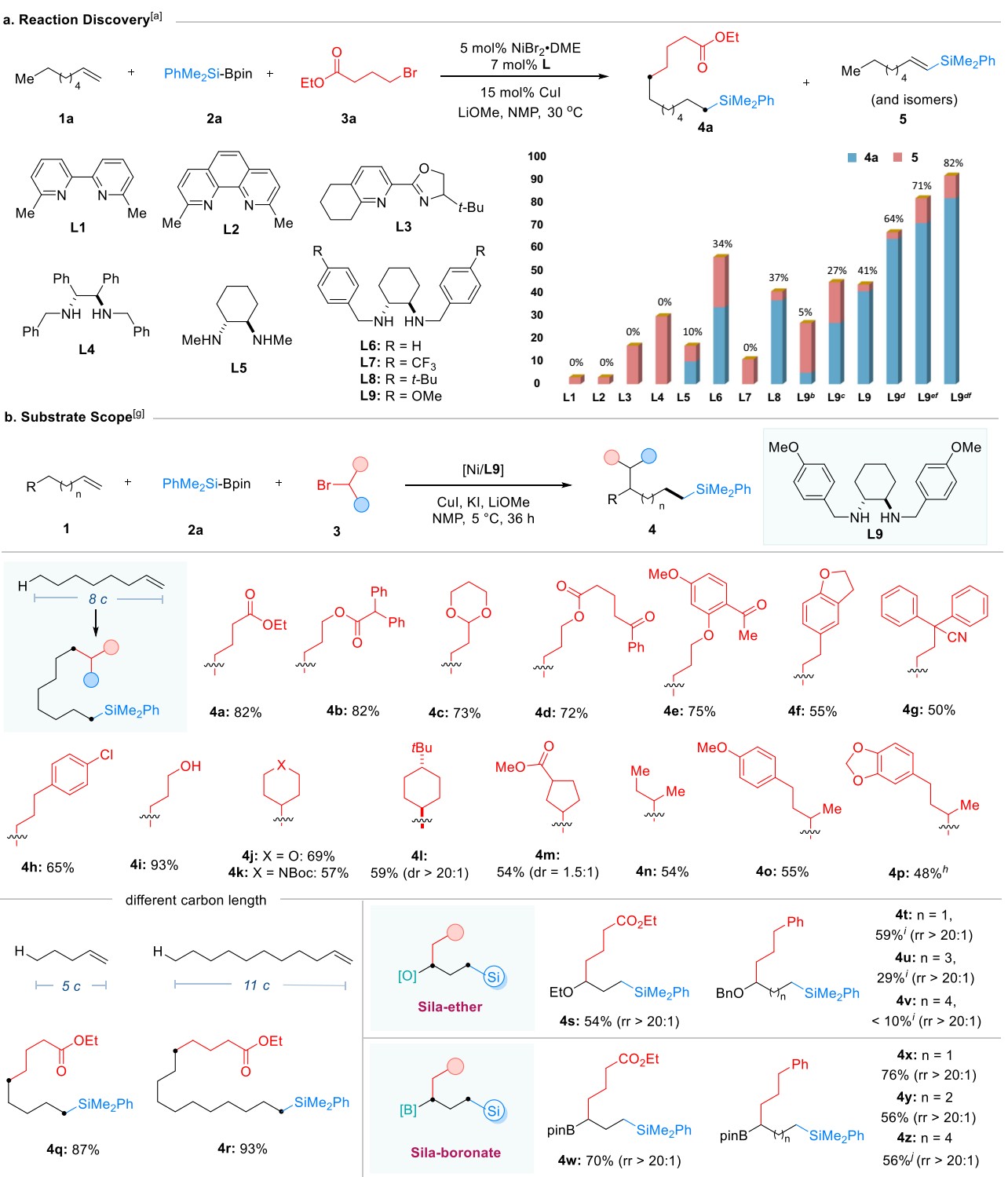

**Fig. 2 | Reaction development and substrate scope. a** Reaction discovery; **b** Substrate scope. [a]Reaction conditions: NiBr$_2$·DME (5 mol%), **L** (7 mol%), CuI (15 mol%), **1a** (0.4 mmol, 1.0 equiv), **2a** (0.8 mmol, 2.0 equiv), **3a** (0.6 mmol, 1.5 equiv), LiOMe (0.8 mmol, 2.0 equiv), NMP (2 mL), 30 °C, 36 h; [b]50 °C; [c]40 °C; [d]5 °C; [e]10 °C; [f]Addition of KI (0.5 equiv); [g]Reaction conditions: NiBr$_2$·DME (5 mol%), **L9**

(7 mol%), CuI (15 mol%), KI (0.5 equiv), **1** (0.4 mmol, 1.0 equiv), **2a** (0.6 mmol, 1.5 equiv), **3** (0.8 mmol, 2.0 equiv), LiOMe (0.8 mmol, 2.0 equiv), NMP (2 mL), 5 °C, 36 h; [h]**L8** instead of **L9**, KI (0.2 equiv); [i]30 °C, 36 h; [j]Isolated yield of the corresponding alcohol after oxidation.

*trans*−1,2-diaminocyclohexane backbone (**L5**-**L9**) were able to furnish a 1,8-regioselective silylalkylation product **4a**, with **L9** providing the optimal performance. The efficiency was further improved by the addition of KI, which was likely due to the coordination of the iodide anion inhibiting β-hydride

elimination[48]. Further study found that the reactivity was highly sensitive to temperature: increasing the reaction temperature to 50 °C resulted in a very poor yield; while decreasing to 5 °C, the yield was improved to 82% with exclusive 1,8-regioselectivity, and this excellent terminal selectivity is kinetically controlled[49].

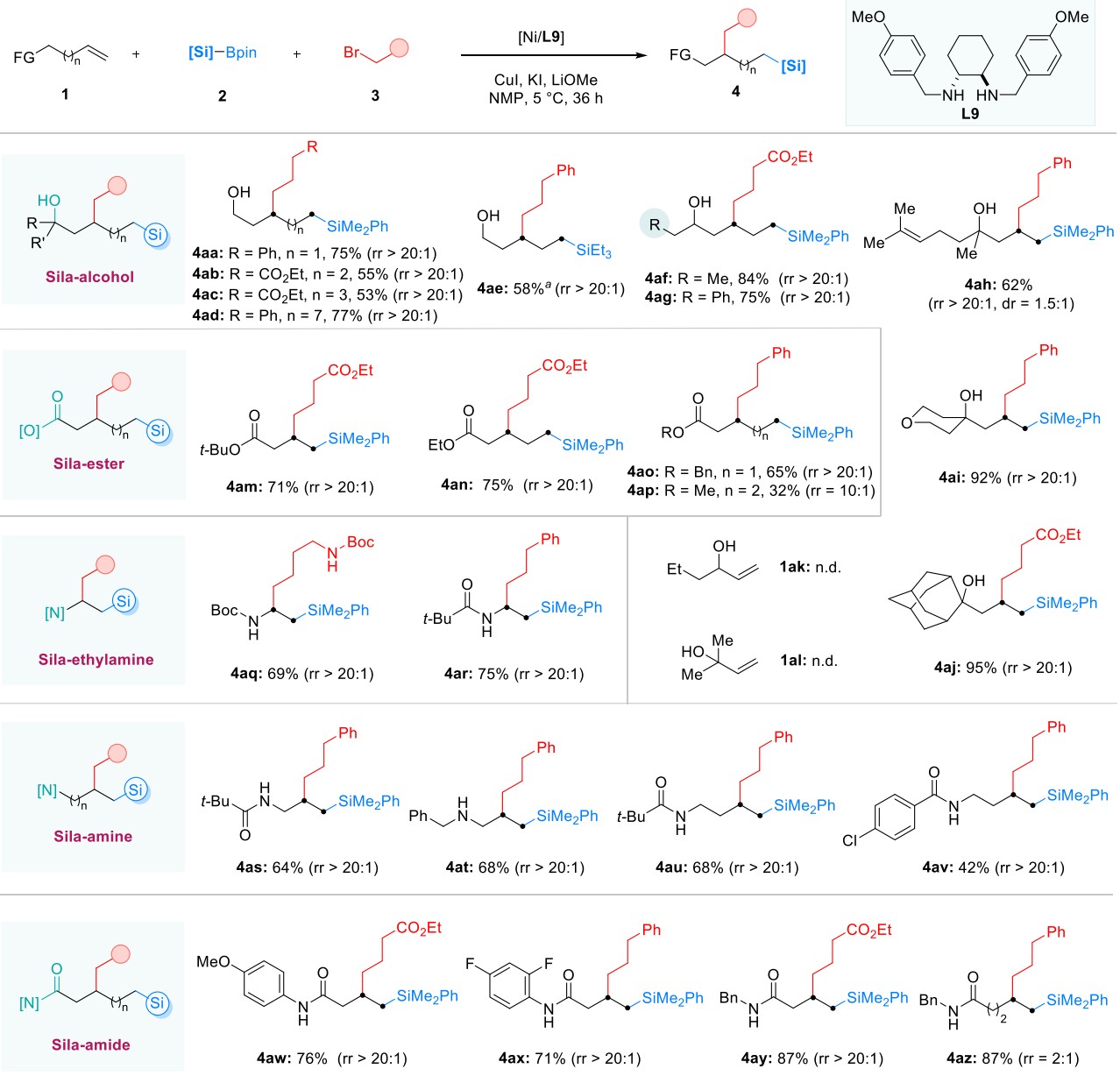

**Fig. 3 | Scope of functionalized alkenes.** Reaction conditions: NiBr$_2$·DME (5 mol%), **L9** (7 mol%), CuI (15 mol%), KI (0.5 equiv), **1** (0.4 mmol, 1.0 equiv), **2** (0.6 mmol, 1.5 equiv), **3** (0.8 mmol, 2.0 equiv), LiOMe (0.8 mmol, 2.0 equiv), NMP (2 mL), 5 °C, 36 h; [a]30 °C, 20 h.

Notably, although the silyl group can also direct the formation of its α-metal complex[50–53], no 1,1-regioselective isomeric product was detected with these conditions.

## Substrate scope

After the optimal reaction conditions were obtained, the generality of this Ni-catalyzed alkene silylalkylation reaction was explored. The scope of the alkyl electrophile was assessed first in combination with 1-octene (**1a**). As shown in Fig. 2b, an array of primary alkyl bromides bearing diverse substituents, including esters (**4a-4c**), ketyl (**4d**), ketones (**4e** and **4 f**), alkenylarene (**4h**), cyano (**4i**), alcohol (**4j**) and aryl chloride (**4k**), participated in this reaction to transfer the corresponding organosilicons in moderate to good yields with exquisite regioselectivity. Moreover, both cyclic (**4l-4o**) and linear (**4p-4r**) secondary bromides could also serve as successful coupling partners to deliver the corresponding products under identical reaction conditions. Both shortening the carbon length to five and extending the

carbon length to eleven afforded the head-tail addition products with an identical level of reactivity (**4q** and **4r**). Moreover, migratory 1,3-regioselective silylalkylation products, generated from thermodynamically favored alkyl intermediates, were isolated from allyl ether and allyl boronate[54–57] (**4s** and **4w**). When extending the carbon length of alkenyl boronate to four or six, the corresponding migratory 1,n-addition products were obtained with only a slight decrease in yields (**4y** and **4z**). However, an increase in chain length led to deleterious reactivity for alkenyl ethers (**4u** and **4v**), possibly due to the influence of the weak coordination of ethers.

To investigate the scope of the alkene component, a library of alkenes containing various functional groups was studied (Fig. 3). Primary (**4aa-4ae**), secondary (**4af** and **4ag**), and tertiary (**4ah-4aj**) alkenyl alcohols reacted smoothly under the Ni-catalyzed system, producing the corresponding sila-alcohols in moderate to excellent yields (53%–95%) and with excellent regioselectivity. Even when the carbon chain length extended to seven or eleven (**4ac** and **4ad**), the

**Fig. 4 | Extended scope and synthetic applications. a** Switchable regioselectivity (1,2-addition vs. 1,4-addition); **b** Synthesis of cyclic sila-ethers; **c** Synthesis of silyl-substituted saturated aza-heterocycles; **d** Concise synthesis of alpha-lipoic acid.

formation of carbon-carbon bonds at the γ-position of the alcohols was uniformly observed in these reactions. Notably, this observation is inconsistent with a recent alcohol-directed alkene difunctionalization reaction[58], wherein 1,2-regioselectivity was uniformly obtained. Regretfully, allyl alcohols did not provide the corresponding silylation

products under the current conditions (**1ak** and **1al**). Comparing with the reactions with alkenyl ethers, we attribute this distinction to the stronger coordination ability of hydroxyl groups with metals compared to alkoxy groups, as demonstrated in a recent work by the Yu group[59]. In addition, the success could also extend to alkenyl esters,

and the corresponding sila-esters (**4am-4ap**) were obtained in good yields with carbon-carbon formation exclusively at the β-position to the ester groups[60]. These results indicate that the involvement of five-membered nickelacycle intermediates in the reactions with alkenyl alcohols or alkenyl esters. Replacing the carboxylic acid group of amino acids with a silyl group can alter their biological activity[61], making the synthesis of amino acid mimics highly valuable in medicinal chemistry. A series of amino acid mimics (**4aq-4av**) were efficiently prepared from vinyl or alkenyl amines. The transformation was also tolerant of various protecting groups, such as *tert*-butyloxy carbonyl (Boc), acyl (Ac), and benzyl (Bz). It is noteworthy that both the carbonyl and the nitrogen atom of amide groups can serve as directing groups[62,63] in these mild reaction conditions (comparing **4as** with **4au** and **4av**). Moreover, 3-butenamides afforded exclusive 1,2-addition products (**4aw-4ay**), but a poor site-selectivity was obtained when extending the chain length (**4az**)[64,65]. Comparing the reactions with alkenyl alcohols or alkenyl esters, these results indicate that the relatively stronger coordination ability of amide groups enables carbon-carbon formation from both five- and six-membered nickelacycle

intermediates in this catalytic system[66,67]. Based on the above findings, regioselectivity-switchable reactions were achieved from homoallyl alcohol (Fig. 4a). In addition, endocyclic alkenes were able to deliver the corresponding cyclic sila-ethers (**4bc** and **4bd**) in good yields with excellent stereochemistry, and a cyclic sila-ether (**6**) was efficiently synthesized by treating the product **4be** with HCl gas (Fig. 4b).

## Synthetic applications

The synthetic potential of this protocol was further demonstrated with downstream transformations of representative products. The first set of transformations involved the preparation of several silyl-substituted saturated aza-heterocycles, which are important pharmacophores ubiquitous in biologically relevant molecules[68] (Fig. 4c). For instance, 2-substituted pyrrolidine (**7**) was prepared from **4bf**, 3-substituted piperidine (**8**) from **4bg**, and the Coniine[69] analog 2-substituted piperidine (**9**) was obtained from **4bh**. Finally, the synthetic value of this methodology was highlighted by simplifying the synthesis of a drug molecule (Fig. 4d). Alpha-lipoic acid (**11**) is a drug with numerous therapeutic functions, including the treatment of rheumatoid arthritis

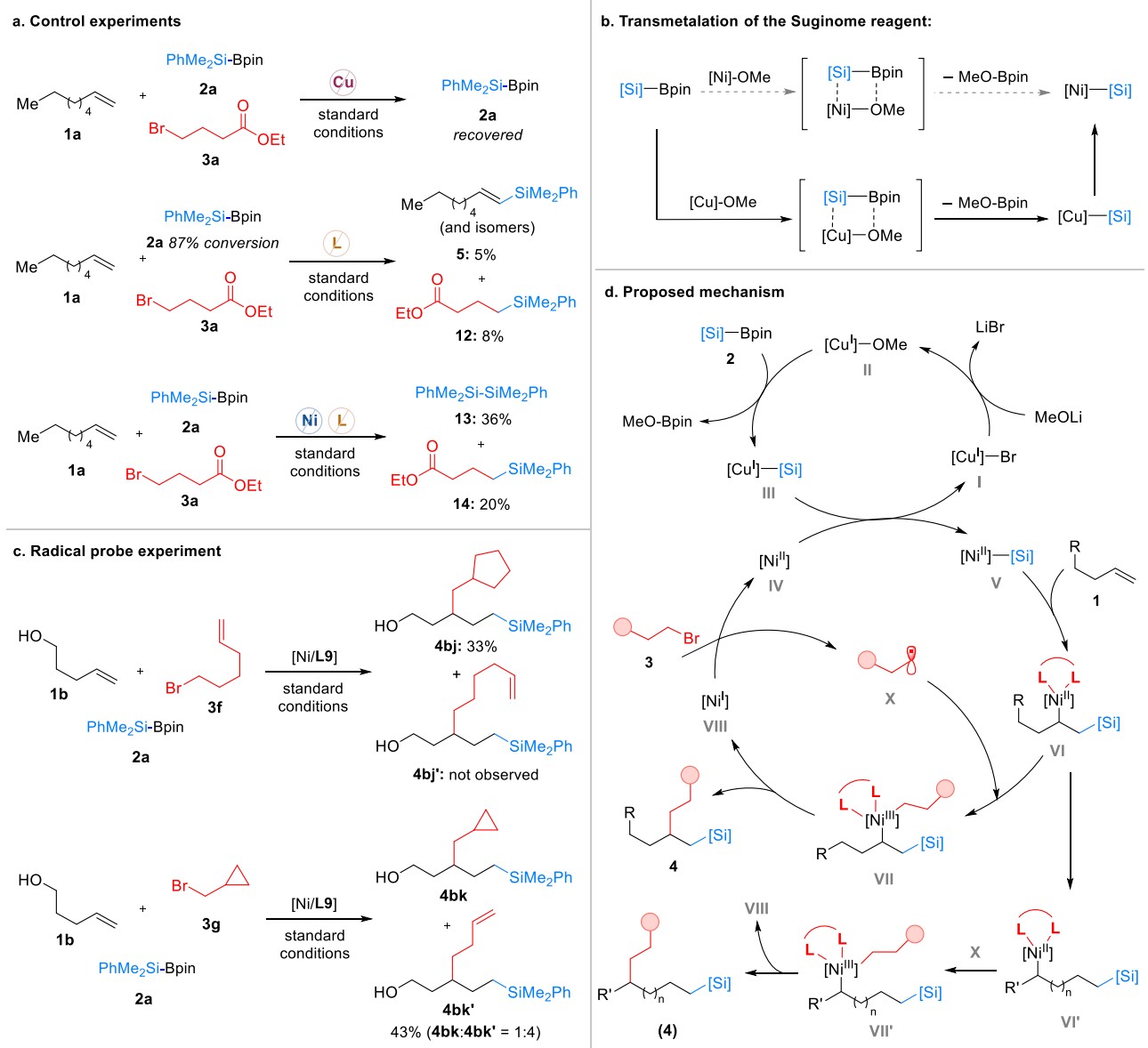

**Fig. 5 | Mechanistic investigations and proposed catalytic cycle. a** Control experiments; **b** Transmetalation of the Suginome reagent; **c** Radical probe experiment; **d** Proposed mechanism.

and Alzheimer's disease[70], and its previous synthetic routes required at least nine steps[71,72]. With this chemistry, a concise route toward alpha-lipoic acid was developed, and the new protocol only requires five steps from inexpensive commodity chemicals.

## Mechanistic studies

Control experiments were carried out to determine the role of the nickel salt, copper salt, and ligand in the three-component reaction (Fig. 5a). The silicon reagent **2a** was recovered from the reaction without copper, indicating that **2a** can not directly react with nickel salt and the copper salt plays an important role in the transmetallation of the silicon reagent to nickel[73,74] (Fig. 5b). Without a ligand, only C–Si bond formation products were detected, revealing the ligand's crucial role in the formation of C–C bond. To shed light on the C–C bond formation step, radical probe experiments were conducted (Fig. 5c). The reaction with 6-bromo-1-hexene (**3f**) produced a cyclization product **4bj** and a ring-opening product **4bk'** was dictated in the reaction with (bromomethyl)cyclopropane (**3g**), suggesting that the generation of alkyl radicals from the alkyl bromides is likely involved in the C–C bond formation step. Based on the results obtained from the experiments and previous studies[39], a catalytic cycle for this bimetallic cooperative catalyzed reaction is proposed. As shown in Fig. 5d, the reaction begins with the transmetallation of the Sugimone reagent (**2**) with a copper salt (**II**) to form a Cu-[Si] species (**III**), which further transmetallates with a nickel salt to generate the Ni-[Si] species (**V**). Olefin migratory insertion is followed to furnish an alkyl-Ni(II) species (**VI**). When an alkene with a neighboring chelating group is used, this species then captures an alkyl radical (**X**) to produce a Ni(III) species (**VII**). Finally, reductive elimination delivers a 1,2-silylalkylation product (**4**) and a Ni(I) species (**VIII**). The intermediate **VIII** reacts with the alkyl halide (**3**) to generate the alkyl radical (**X**) and the Ni(II) catalyst (**IV**) and to close the cycle. When an aliphatic olefin or an alkene with a remote chelating group is used, rapid chain-walking occurs on the intermediate **VI**, leading to generate a thermodynamically stable species (**VI'**). Then the species undergoes radical addition and reductive elimination (via **VII'**) to deliver the corresponding migratory product.

## Discussion

In conclusion, a highly regioselective silylalkylation of non-activated terminal alkenes has been developed, producing an efficient approach to sp³-rich organosilicon building blocks. The reaction regioselectivity is determined by the alkene and the choice of bisnitrogen-ligated nickel catalyst is crucial to the success of these transformations, leading to mild reaction conditions, exquisite site-selectivity, wide substrate scope, and exceptional functionality tolerance. Follow-up transformations of selected products demonstrate the synthetic potential of this chemistry, including the preparation of important pharmacophores and the streamlining of a synthetic route for a drug molecule. Overall, this study has the potential to significantly impact medicinal chemistry research.

## Methods

### Model procedure

Under $N_2$ atmosphere in a glove box, an oven-dried 10 mL reaction tube equipped with a magnetic stir bar and sealed with a rubber stopper was used. Sequentially, NiBr₂·DME (6.2 mg, 0.02 mmol, 5 mol %), **L9** (10.0 mg, 0.028 mmol, 7 mol%), LiOMe (30.4 mg, 0.8 mmol, 2.0 equiv), CuI (11.4 mg, 0.06 mmol, 15 mol%), KI (33.0 mg, 0.20 mmol, 0.5 equiv) and PhMe₂SiBpin (220.8 mg, 0.8 mmol, 2.0 equiv) were added. Subsequently, anhydrous NMP (1 mL), alkenes (0.4 mmol, 1.0 equiv), alkyl bromide (0.6 mmol, 1.5 equiv) and anhydrous NMP (1 mL) were added in sequence, and the mixture was stirred at room temperature. After stirring at 5 °C for 36 h (or stirring at 30 °C for 24 h), the resulting mixture was quenched with water (2 mL) and further

diluted with ethyl acetate (3 mL). The mixture was then extracted with ethyl acetate, and the combined organic layers were dried over anhydrous $Na_2SO_4$, filtered, and concentrated under a vacuum. The crude material was separated on a silica gel column to afford the desired product.

## Data availability

The authors declare that all data supporting the findings of this study are available within the article and Supplementary Information files and all data are also available from the corresponding author upon request.

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

## Acknowledgements

We acknowledge the National Natural Science Foundation of China (22122107), the Fundamental Research Funds for Central Universities (2042022kf1023), the Large Instruments and Equipment Open Sharing Subsidies of Wuhan University (LF20221537), and Wuhan University for financial support.

## Author contributions

Y.G. designed and supervised the project. D.C. developed the catalytic method. D.C., R.Y., and Y.Y. supported the design and performance of synthetic experiments. All the authors were involved in the analysis of the results and discussions of the project.

## Competing interests

The authors declare no competing interests.
