## [Peer Review File · Nature Communications]

Ligand-Modulated Nickel-Catalyzed Regioselective Silylalkylation of AlkenesReviewers' Comments:

Reviewer #1:

Remarks to the Author:

Yin reports a Ni and Cu catalyzed silyl alkylation reaction. Depending on the substrate used, the regioselectivity can vary. However, this is quite predictable based on the presence or absence of directing groups. Silyl functionalization reactions are less well developed compared to related borylation process. However, the silyl reaction is valuable due to the utility of the C-Si bond and potential for biological study. The substrate scope is well investigated. In each case it is clear why the product is made (fig 3). The mechanism studies are also informative.

Overall, I am highly supportive of publication because this method represents a new way to functionalize alkenes in a predictable manner and the products are useful.

Minor comment:

The Yin group has previously reported a Silylarylation (ref 69). I understand that what the authors have done here is excellent, but the prior work should be discussed earlier and likely highlighted in the first scheme.

Reviewer #2:

Remarks to the Author:

Yin and co-workers report an interesting three component carbo-silylation of alkenes using silyl-boranes, alkenes, and alkyl bromides, using nickel and copper co-catalyst. The reaction proceeds via chain walking after the initial silylation of the alkene, and the product outcome depends on the nature of the terminal functional group opposite the alkene. This is an interesting and useful reaction, and overall this is a very high quality paper worthy of publication in Nature Communications. However, there are a few minor concerns that should be addressed before publication.

- 1) In Figure 2, 4n, 4o and 4p are incorrectly (I believe) reported to proceed with "dr". This does not make sense given the products, and appears to be a typo in the text.
- 2) For several of the compounds, some detail (either in the SI or in the text) should be provided to describe how the regioisomer was determined. Particularly for compounds such as 4s-4v, the simple proton NMR's that are provided are not enough to conclusively determine product identity. 2D NMR and analysis should be provided.
- 3) In Fig 3, the inclusion of the "lysine mimic" is misleading as the products made do not map onto it directly. I think this should be removed from the figure.
- 4) An example of the failure of an allylic alcohol should be included in Figure 3, not just mentioned in the text and SI.
- 5) The text refers to the reported synthesis of alpha-lipoic acid as a four-step synthesis... I count five steps, carried out in four-pots. This should be corrected.
- 6) The SI reports that proton NMR were referenced against Me₄Si, but it does not appear that Me₄Si is in all of the spectra. Please clarify.
- 7) The General Procedure is too vague. It refers to solids being added to the reaction tube under a nitrogen atmosphere, but does not say how this was accomplished. It also mentioned addition of alkene, aryl bromide, silyl-borane, and solvent but does not say how they were added (I assume syringe?). What about solids of these compounds? Were reagents or solvents degassed? What temperature were the additions performed at? The reaction was run at 5 C... but were the additions at rt? There are far too few details provided.
- 8) The spectra should be labeled with experiment type. Non-experts may have trouble identifying silicon and boron NMR.

Reviewer #3:

Remarks to the Author:

This paper describes a three-component alkylsilylation reaction of alkenes by a cooperative catalytic system derived from nickel and copper. The authors have previously reported primary-alkylborylation and arylsilylation reactions of alkenes under similar reaction conditions, and this study can be considered as an extension of those efforts. However, a notable aspect is the expanded variation of alkenes with different remote chelating groups, which could potentially provide a useful method for synthesizing various functionally functionalized alkylsilanes in organic synthesis. The reviewer may support its publication in Nature Comm, provided that the following points are adequately addressed.

1. Regarding the remote chelating groups, several questions arise, requiring further discussion. For instance, why esters provide excellent regio-selectivity, but amides do not yield similarly favorable results (4ag vs 4aq)? Additionally, to what extent can these functional groups, including hydroxyl groups, function remotely? Why is the remote chelating effect not observed for N-acylamines (4al and 4am)? In the absence of a remote chelating group, how does the process manage to extend all the way to the terminal position? How do alkoxy and boryl groups act as remote chelating groups, and how would their effects differ if they were placed further away? Is there a distinction between hydroxyl and alkoxy groups in this context?

2. Although chiral ligands are employed, is there no enantioselectivity observed? If facial selectivity could be induced during the silylnickelation step, enantioselectivity could potentially be achieved even with a chain walking process.

Response to Reviewers' Comments

Reviewer: 1

Comments:

Yin reports a Ni and Cu catalyzed silyl alkylation reaction. Depending on the substrate used, the regioselectivity can vary. However, this is quite predictable based on the presence or absence of directing groups. Silyl functionalization reactions are less well developed compared to related borylation process. However, the silyl reaction is valuable due to the utility of the C-Si bond and potential for biological study. The substrate scope is well investigated. In each case it is clear why the product is made (fig 3). The mechanism studies are also informative.

Overall, I am highly supportive of publication because this method represents a new way to functionalize alkenes in a predictable manner and the products are useful.

Our Response: We deeply appreciate the reviewer for his/her positive comments and are immensely grateful for his/her recommendation and support for the publication of our paper in Nature Communications.

Minor comment:

The Yin group has previously reported a Silylarylation (ref 69). I understand that what the authors have done here is excellent, but the prior work should be discussed earlier and likely highlighted in the first scheme.

Our Response: We thank the reviewer for this suggestion. Revisions have been made as suggested in the revised manuscript.

Reviewer: 2

Comments:

Yin and co-workers report an interesting three component carbo-silylation of alkenes using silyl-boranes, alkenes, and alkyl bromides, using nickel and copper co-catalyst. The reaction proceeds via chain walking after the initial silylation of the alkene, and the product outcome depends on the nature of the terminal functional group opposite the alkene. This is an interesting and useful reaction, and overall this is a very high quality paper worthy of publication in Nature Communications. However, there are a few minor concerns that should be addressed before publication.

Our Response: We are deeply grateful for the reviewer's high praise of our work and their agreement to have it published in Nature Communications.

Minor comments:

1. In Figure 2, **4n**, **4o** and **4p** are incorrectly (I believe) reported to proceed with “dr”. This does not make sense given the products, and appears to be a typo in the text.

Our response: We express our sincere appreciation to the reviewer for identifying this mistake in our manuscript. We have made correction as suggested in the revised manuscript.

2. For several of the compounds, some detail (either in the SI or in the text) should be provided to describe how the regioisomer was determined. Particularly for compounds such as **4s-4v**, the simple proton NMR's that are provided are not enough to conclusively determine product identify. 2D NMR and analysis should be provided.

Our response: We sincerely appreciate the reviewer for this suggestion. In fact, the structures of several compounds have been identified through oxidation; please see SI Page 36 for details. Additionally, the structures of compounds **4s-4v** have been careful assigned by assistance of NMR spectra; please see the revised SI for comprehensive details.

3. In Fig 3, the inclusion of the “lysine mimic” is misleading as the products made do not map onto it directly. I think this should be remove from the figure.

Our response: We thank the reviewer for this constructive suggestion. Accordingly, it has been removed as suggested in the revised manuscript.

4. An example of the failure of an allylic alcohol should be included in Figure 3, not just motioned in the text and SI.

Our response: We have included an example of an allylic alcohol in the revised manuscript. Please see the Fig 3. We sincerely appreciate the reviewer for their comment.

5. The text refers to the reported synthesis of alpha-lipoic acid as a four-step synthesis... I count five steps, carried out in four-pots. This should be corrected.

Our response: We do apologize for the inaccuracy in our previous descriptions. We have carefully addressed and corrected this error in the revised manuscript.

6. The SI reports that proton NMR were referenced against Me₄Si, but it does not appear that Me₄Si is in all of the spectra. Please clarify.

Our response: We do thank the reviewer for pointing out this. We have taken this

suggestion into account and made the appropriate correction in the revised SI.

7. The General Procedure is too vague. It refers to solids being added to the reaction tube under a nitrogen atmosphere, but does not say how this was accomplished. It also mentioned addition of alkene, aryl bromide, silyl-borane, and solvent but does not say how they were added (I assume syringe?). What about solids of these compounds? Were reagents or solvents degassed? What temperature were the additions performed at? The reaction was run at 5 C... but were the addition at rt? There are far too few details provided.

Our response: We sincerely apologize for the inaccuracy descriptions in the general procedure. We have carefully addressed some errors and added more details in the revised SI. Finally, we genuinely appreciate the reviewer for bringing these issues to our attention and helping us improve the accuracy of our work.

8. The spectra should be label with experiment type. Non-experts may have trouble identifying silicon and boron NMR.

Our response: We do thank the reviewer for pointing out this. Accordingly, we have labeled the type of spectra (^1H 、 ^{13}C 、 ^{29}Si and ^{11}B) in the SI.

Reviewer: 3

Comments:

This paper describes a three-component alkylsilylation reaction of alkenes by a cooperative catalytic system derived from nickel and copper. The authors have previously reported primary-alkylborylation and arylsilylation reactions of alkenes under similar reaction conditions, and this study can be considered as an extension of those efforts. However, a notable aspect is the expanded variation of alkenes with different remote chelating groups, which could potentially provide a useful method for synthesizing various functionally functionalized alkylsilanes in organic synthesis. The reviewer may support its publication in Nature Comm, provided that the following points are adequately addressed.

Our response: We wholeheartedly appreciate the reviewer's constructive comments on our work, and we would like to express our sincere gratitude for granting us an opportunity to strengthen our paper, making it more suitable for publication in Nature Communications.

Minor comments:

1. Regarding the remote chelating groups, several questions arise, requiring further discussion. For instance, why esters provide excellent regio-selectivity, but amides

do not yield similarly favorable results (**4ag** vs **4aq**)? Additionally, to what extent can these functional groups, including hydroxyl groups, function remotely?

Our response: We greatly appreciate the reviewer for these insightful questions.

Regarding the difference of esters and amides, we speculate that the varied results are likely attributed to the stronger coordination ability of amide groups with metals compared to that of ester groups. In ester substrates, only a stable five-membered nickelacycle intermediate proved effective in directing carbon-carbon bond formation. Conversely, the stronger coordination of amide groups enables less stable six-membered nickelacycle intermediates to also direct carbon-carbon bond formation in amide substrates.

Regarding to the extent to which these functional groups, including hydroxyl groups, can function remotely, we conducted additional experiments as shown below. From these results, it is evident that extending the chain length to eight or eleven has no adverse effect on the reactivity and regioselectivity of alkenyl alcohols. However, an increase in chain length resulted in a decrease in the yield of alkenyl ester, although its regioselectivity remained good.

The above discussion and results have been added to the revised manuscript.

2. Why is the remote chelating effect not observed for *N*-acylamines (**4al** and **4am**)?

Our response: We thank the reviewer for this good question. We speculate that it is the nitrogen atom of amide coordination to direct the formation of a five-membered nickelacycle intermediates in the reaction of *N*-acylamines (**4al** and **4am**). Amines can function as directing groups was also reported in an Engle's recent work (ref. 63). We have made revision in the revised manuscript.

3. In the absence of a remote chelating group, how does the process manage to extend all the way to the terminal position?

Our response: We greatly appreciate the reviewer's insightful question. We think this terminal selectivity is kinetically controlled selectivity, same as the regioselectivity in migratory hydrofunctionalization reactions (for a review: ref. 49).

4. How do alkoxy and boryl groups act as remote chelating groups and how would their effects differ if they were placed further away? Is there a distinction between hydroxyl and alkoxy groups in this context?

Our response: We appreciate the reviewer for these good questions. Firstly, the empty *p* orbital of the boron group can provide an electron-withdrawing factor to stabilize the C-Ni bond, serving as a driving force for chain-walking in reactions involving alkenyl boronates. In the case of alkoxy groups, oxygen can function as an electron-withdrawing factor, and the lone pair on oxygen can serve as a coordinating factor to stabilize the ortho-alkyl nickel species. This stabilization acts as a thermodynamic driving force for chain-walking in the reactions involving alkenyl ethers.

Secondly, in response to the reviewer's comments, we have examined several alkenes with longer carbon chains. As shown below, longer carbon chains do not negatively impact on the yield and regioselectivity of alkenyl boronates. However, an increase in chain length led to a decrease in the yield in the case of alkenyl ethers, although the regioselectivity remained good. This is likely due to the influence of the weak coordination of ethers.

Thirdly, hydroxyl groups and alkoxy groups yielded distinct results in this reaction. We attribute this distinction to the stronger coordination ability of hydroxyl groups with metals compared to alkoxy groups, with the latter exhibiting weaker coordination. This distinction probably arises from the fact that the coordination of hydroxyl groups with metals enhances the acidity of hydrogen, potentially forming hydrogen bonds that further strengthen their coordination ability, as demonstrated in a recent work by the Yu group (ref. 59).

The above results and discussion have been added to the revised manuscript.

5. Although chiral ligands are employed, is there no enantioselectivity observed? If facial selectivity could be induced during the silylnickelation step, enantioselectivity could potentially be achieved even with a chain walking process.

Our response: We sincerely appreciate the reviewer for this question. As shown below, when chiral (*R,R*)-**L9** was used, the products **4ak** and **4w** exhibited almost no enantiomeric excess (ee). This result indicates that the current ligand structure is not suitable for achieving high enantioselectivity control. Since diverse functional group can serve as directing group, we believe that it is still difficult to find a suitable chiral ligand to achieve enantioselectivity control for a broad range of alkenes.

4aa: 75%, ee = 2%

4au: 68%, ee = 7%

Racemic **4aa**

Enantioenriched **4aa**

Racemic **4au**

Enantioenriched **4au**

Reviewers' Comments:

Reviewer #2:

Remarks to the Author:

The corrections the authors have provided satisfy all of my concerns. I recommend publication.

Reviewer #3:

Remarks to the Author:

The revised manuscript has addressed most of issues raised by the previous reviewers and thus can be accepted for publication.

Response to Reviewers' Comments

Reviewer #2:

The corrections the authors have provided satisfy all of my concerns. I recommend publication.

Our Response: We deeply appreciate the reviewer for their constructive suggestions and positive comments that have significantly improved our research. We are delighted to see that our response has met the satisfaction of the reviewers. And we are profoundly grateful for the reviewer's agreement to have our work published in Nature Communications.

Reviewer #3:

The revised manuscript has addressed most of issues raised by the previous reviewers and thus can be accepted for publication.

Our Response: We sincerely thank the reviewer for acknowledging that our studies effectively address the majority of the issues raised by the previous reviewers. Furthermore, we are profoundly grateful for his/her agreement to have our work published in Nature Communications.